# Latent Profile Analysis of Children’s Moral Character and the Classing Effect on Bullying in Rural China

**DOI:** 10.3390/ijerph191811285

**Published:** 2022-09-08

**Authors:** Ruiping Zhang, Linlin Gao, Lan Cheng, Ping Ren

**Affiliations:** 1School of Education, Zhengzhou University, Zhengzhou 450001, China; 2Collaborative Innovation Centre of Assessment toward Basic Education Quality, Beijing Normal University, Beijing100875, China

**Keywords:** moral character, bullying/bullied behavior, left-behind children, latent profile analysis

## Abstract

Moral character is the key component of positive youth development. However, few studies have examined children’s moral character and the association with bullying and bullied behavior. Guided by the framework of positive psychology, this study aimed to investigate the association of moral character with bullying and bullied behavior among children in rural China and whether the association differed between left-behind children (LBC) and non-left-behind children (NLBC). A total of 723 children (aged 11–16 years) in rural China completed standard questionnaires that contained six specific character traits and bullying/bullied behavior. Latent profile analysis revealed that children’s moral character was divided into three classes (i.e., low-character class, average-character class, and high-character class). Compared with children in low-character and average-character classes, children in the high-character class had the lowest bullying and bullied behavior. Children in the low-character class were those at greater risk of bullied behavior. The association of the latent character classes with bullied behavior differed between LBC and NLBC. These findings highlight the urgent need for character-based and targeted interventions to prevent children’s bullying and bullied behavior.

## 1. Introduction

With the deepening of economic reform and opening up, China’s modernization and urbanization have been accelerating, and the surplus labor force in rural areas has been transferred to big cities for better job opportunities. However, due to the restrictions of China’s household registration system and financial constraints, migrant parents have to leave their children behind in their hometown to be looked after by others [1]. LBC have become a particular youth population in China that deserve our earnest attention [2]. Previous literature has mainly been based on the deficit model and focused on the problems and potential risks that LBC face [3,4]. For example, compared with non-migrant children, LBC have an increased risk of depression, anxiety, suicidal ideation, conduct disorder, substance use, and stunting [5]. Some researchers argue that if a child had developmental deficits in childhood that were not directed and corrected during their critical periods of development, those deficits would be difficult to change in adulthood [6,7]. Different from the deficit model, Positive Youth Development (PYD) emphasized the developmental potential of adolescents themselves rather than incompetence [8]. PYD was primarily concerned with three areas of children’s development: the nature of the child; the interaction between the child and the community; and moral growth [6]. Among these three areas, the role of moral character was prominent, which may be more uniformly and globally associated with positive outcomes early in development [9,10].

## 2. Background

### 2.1. Moral Character

In this study, moral character is defined as a person’s characteristic patterns of thinking, feeling, and behavior related to moral/ethical and unethical behavior [11]. Cohen et al. [11] assumed that moral character was not a single dimension of personality but a multifaceted structure composed of broad and narrow traits. Grounded in positive psychology, Peterson and Seligman [12] discovered 6 core virtues and 24 character strengths. However, these character strengths were wide ranging. Which traits should be considered as moral character traits is ambiguous [9]. Cohen et al. [11] used latent profile analysis to divide adult employees’ moral character into three classes: low-moral-character class, average-moral-character class, and high-moral-character class. However, these results cannot be directly applied to children and adolescents.

Although the structure of moral character among children is unclear, previous studies have indicated gender and age differences in character. Women were found to have higher moral character than men [11]. Shubert et al. [9] found that global character strength was more evident among elementary school-aged children than among middle school students. However, the participants of the above studies were mainly from western cultures. There has been a relative absence of studies on children’s moral character in eastern cultures, especially among LBC. Although limited, some studies have focused on specific attributes of character in LBC. For example, LBC may develop better gratitude and conscientiousness than NLBC, while may also become aggressive and indifferent [13].

### 2.2. The Association of Moral Character with Children’s Bullying and Bullied Behavior

Bullying is repeated offensive behavior that deliberately hurts or harasses weak people or groups, which can lead to detrimental impacts; for instance, bullying can affect the general well-being, academic achievement, and social functioning of its victims [14,15]. Furthermore, mental problems related to childhood bullying may continue into late adolescence and even adulthood [16]. Recently, bullying among school-aged children in China has become a severe problem [15]. The prevalence of self-reported school bullying in mainland China ranged from 2 percent to 34 percent [17]. It is important to examine school bullying in China.

According to the relational developmental systems theory, moral character involves linking across time and place to provide mutually positive benefits to both self and others [18]. Enhancing children’s character could benefit both individuals and civil society [19]. Complementary to the relational developmental systems theory is the dyadic agent–patient model of morality, which proposes that harmful acts are committed by moral agents and these acts cause suffering to moral patients [11]. Previous studies have indicated the association of some specific character traits with bullying and bullied behavior. For example, lower levels of honesty–humility and conscientiousness were associated with both bullying and bullied behavior, while empathy was negatively associated with only bullying behavior [20,21]. Character strengths have been found to be related to a series of important life outcomes, such as life satisfaction, academic achievement, work performance, relationships, and health-related behaviors [22].

In addition, employees with low moral character were more likely to engage in harmful work behaviors and delinquent behavior than high-moral-character employees [11]. More importantly, children with bullying behavior had a lower level of initial moral character than their counterparts [23]. However, few studies have examined the relationship between personality traits and bullying in LBC and even fewer have investigated the relationship between moral character and LBC’s bullying.

### 2.3. The Possible Moderating Effect of Left-behind Status

Previous studies mostly regarded left-behind status (LBC vs. NLBC) as an independent variable or a controlled variable, and seldom examined the moderating effect of left-behind status [24]. Little work has investigated the combined effects of moral character and left-behind status on children’s bullying and bullied behavior. Researchers found that LBC with high social support had better social adaptation than NLBC with high social support [25]. Thus, there are reasons to assume that the association of moral character with children’s bullying and bullied behavior may differ between LBC and NLBC. On the one hand, left-behind status was associated with bullying [14] and bullied behavior [15]. Parental absence was more likely to increase the risk of child victimization and accidental injury [26]. On the other hand, LBC may lack empathy and become indifferent due to a lack of parental supervision and support [17]. However, Wen et al. [27] found that parental migration might be not a risk factor for youth development in terms of at least character. Nguyen [28] indicated that the negative effect on children tends to be higher for long-term parental migration than for short-term parental migration. These findings may provide a clue for the possible moderating effect of left-behind status.

### 2.4. The Present Study

According to PYD, LBC have the potential for good development. Character is a core element of PYD [10], and the relational developmental systems theory indicates moral character could provide mutually positive benefits to both self and others [18]. In accordance with these perspectives, the criterion variables used in this study are children’s bullying and bullied behavior. However, only a few studies have examined specific character traits in LBC and the associations with bullying and bullied behavior. Until now, no research has ever directly examined the possibility that the association of moral character with bullying and bullied behavior differs between LBC and NLBC.

We selected some specific traits of moral character by searching for available literature on LBC and conducted latent profile analysis (LPA) to determine which measures best-distinguished individuals with low moral character from those with high moral character. This person-centered method could capture all information at the individual level [29]. Based on the above theoretical foundation and empirical studies, this study firstly explored the potential latent classes of children’s moral character. Next, we explored the relationship of moral character with children’s bullying and bullied behavior. Finally, we investigated whether left-behind status moderated the association of moral character with children’s bullying and bullied behavior. We postulated that children’s latent moral character could be divided into three classes (i.e., low-character class, average-character class, and high-character class), and children’s moral character was correlated with bullying and bullied behavior, and left-behind status could moderate the association of moral character with children’s bullying and bullied behavior.

## 3. Method

### 3.1. Participants and Procedures

Data were collected from 8 rural primary and middle schools in Henan province, China, including 4 primary schools and 4 middle schools. In 2018, there were approximately 699,000 LBC in Henan Province, accounting for 10.1% of the total number of LBC in the country. These eight schools were all from economically underdeveloped areas of Henan Province, a region of midland China with a substantial proportion of migrant labor. The current data collected are therefore rather representative of the general LBC in China. This study defined LBC as those below 16 years of age, with one or both of their parents migrating from rural to urban areas for over six months. The final samples (*N* = 723) included 288 LBC and 435 NLBC. There were 338 boys and 384 girls (1 subject did not report his gender), with an average age of 11.56 years (range = 11–16, *SD* = 1.78). There were more students in the 7th grade (55.04%) than those in the 4th grade (44.96%).

This study was approved by the research ethics committee of our institution. The researchers obtained the informed consent of parents and participants before data collection. The participants were assured that they were free to withdraw and that their responses would be kept confidential. We designed student questionnaire 1, student questionnaire 2, and parent questionnaire to conduct this investigation. Student questionnaire 1 was used to obtain children’s demographic information and character traits. Student questionnaire 2 was used to obtain information on children’s bullying and bullied behavior. In this study, we did not use information reported by parents. The students were asked to complete student questionnaire 1 and student questionnaire 2 in their classroom during different class sessions that lasted approximately 30 min. The researchers explained the requirements and instructions during the survey in classrooms and guided the participants to ensure that they correctly understood the questionnaire.

### 3.2. Materials

#### 3.2.1. Self-Control Scale

Children’s self-control was evaluated using the Chinese version [30] of the Self-control Scale (SCS) [31]. This scale consisted of 13 items (e.g., “I can resist temptation very well.”). Responses were measured on a 5-point Likert scale ranging from 1 (*strongly* disagree) to 5 (*strongly agree*). Higher scores represented better self-control. In the present study, the internal reliability of this scale was 0.73.

#### 3.2.2. Gratitude Scale

Children’s gratitude was evaluated using the Chinese version of the gratitude scale [32,33]. This scale consisted of 6 items (e.g., “I think there’s so much to be thankful for in life.”). Responses were measured on a 6-point Likert scale ranging from 1 (strongly *disagree*) to 6 (*strongly agree*). Higher scores represented higher levels of gratitude. In this study, the internal reliability of the scale was 0.56.

#### 3.2.3. Interpersonal Reactivity Index-C

Children’s empathy was evaluated using the Chinese version [34] of the Interpersonal Reactivity Index-C (IRI-C) [35]. The IRI-C consisted of 22 items (e.g., “I will refer to different opinions before making a decision.”). Responses were measured on a 5-point Likert scale ranging from 1 (*strongly disagree*) to 5 (*strongly agree*). This study assessed empathy using two subscales, i.e., the 7-item empathic concern and 7-item perspective-taking. Higher scores represented higher empathy. In this study, the internal reliability of the scale was 0.65.

#### 3.2.4. NEO-Five Factor Inventory

Children’s conscientiousness was evaluated using the Chinese version of the NEO-Five Factor Inventory (NEO-FFI) [36,37]. The NEO-FFI consisted of 60 items, of which 12 items assessed conscientiousness (e.g., “I do things carefully, and check again after I finish one thing.”). Responses were measured on a 5-point Likert scale ranging from 1 (*not like me at all*) to 5 (*very much like me*). Higher scores represented higher levels of conscientiousness. In this study, the internal reliability of the scale was 0.87.

#### 3.2.5. Kiddie Mach Scale

Children’s Machiavellianism was evaluated using the Chinese version [38] of the Kiddie Mach scale (KMS) [39]. This scale consisted of 16 items (e.g., “Don’t tell anyone the real reason you’re doing something, unless you have a special purpose.”). Responses were measured on a 4-point Likert scale ranging from 1 (*strongly disagree*) to 4 (*strongly agree*). Higher scores represented higher Machiavellianism. In this study, the internal reliability of the scale was 0.46, which was similar to previous findings conducted in Chinese samples [40].

#### 3.2.6. Callous-Unemotional Traits

Children’s uncaring was evaluated with the inventory of Callous-Unemotional Traits (ICU) [41], which consisted of 24 items. In this study, the uncaring subscale was used (8 items, e.g., “I will be frank about how I feel.”). Responses were measured on a 4-point Likert scale ranging from 1 (*strongly disagree*) to 4 (*strongly agree*). Higher scores represented higher uncaring. The internal reliability of the scale was 0.63 in the present study.

#### 3.2.7. Bullying/Bullied Questionnaire

The Chinese version of the Olweus Bullying/Bullied Questionnaire [42,43] was used to evaluate children’s bullying and bullied behavior in the past three months. This questionnaire consisted of 14 items (e.g., “teasing or playing tricks on others.”). Responses were measured on a 5-point Likert scale ranging from 0 (*none*) to 4 (*five times or more*). A higher score represented more bullying/bullied behavior. In this study, the internal reliability of the two dimensions was 0.83 and 0.83, respectively.

### 3.3. Statistical Analysis for LPA

First, a descriptive statistic was used to examine children’s latent moral character. LPA was implemented in Mplus7.0 to distinguish moral character profiles based on Z scores of these six character traits (i.e., self-control, gratitude, empathy, conscientiousness, Machiavellianism, and uncaring). Model fit was based on the Akaike information criterion (AIC), Bayesian information criterion (BIC), Entropy, and Lo–Mendell–Rubin likelihood ratio test (LMR-LRT) [29,44]. Low scores of these indices showed a good fit to the data. Entropy indicated model classification accuracy, and if Entropy was more significant than 0.80, the model classification accuracy exceeds 90%. LMRT was used to compare the model, and a significant value (*p* < 0.001) indicated the k model is better than the k – 1 model. Theoretical interpretability of the classes was considered in comparing models with similarly good fit statistics. Second, mixed regression analysis and multivariate analysis of covariance were used to analyze the association of children’s latent moral character with bullying/bullied behavior and the possibility that this association differs between LBC and NLBC.

## 4. Results

### 4.1. Latent Profile Analysis of Children’s Moral Character

We examined models with up to four latent classes and ultimately selected a three-class model by comparing different models’ interpretability and statistical robustness. Table 1 summarizes the necessary model indices of the LPA results.

Why were three categories chosen as the optimal model? They were the following reasons. First, based on LMRT, the two-class, three-class, and four-class models were acceptable. Second, based on Entropy, the two-class, three-class, and four-class models were acceptable. Third, based on AIC and BIC, the two-class model was rejected. Finally, compared with the four-class model, the class proportion of the three-class model was more appropriate. Thus, the three-class model was chosen as the optimal model.

The three latent classes were depicted in Figure 1. The probabilities of three latent classes were 63.6%, 14.5%, and 21.8%, respectively, which were named average-character class, low-character class, and high-character class. Profile 1 was termed average-character class, which comprised 63.6% of the total sample, and representative participants showed average levels of self-control, gratitude, empathy, conscientiousness, Machiavellianism, and uncaring. Profile 2 was termed low-character class, which comprised 14.5% of the total sample, and representative participants showed low levels of self-control, gratitude, empathy, and conscientiousness and high levels of Machiavellianism and uncaring. Profile 3 was termed high-character class, which comprised 21.8% of the total sample, and representative participants showed high levels of self-control, gratitude, empathy, and conscientiousness and low levels of Machiavellianism and uncaring.

In this study, self-control, gratitude, empathy, conscientiousness, Machiavellianism, and uncaring differentiated high-character class from low-character class by approximately 1.5 standard deviations (*SD*s) or more. Uncaring was the trait that most differentiated low-character class from high-character class.

### 4.2. The Association of Latent Character Classes with Bullying and Bullied Behavior

With latent character classes as the independent variable and school bullying as the dependent variable, a mixed regression model was established in Mplus7.0, after age and gender were controlled for. As seen in Table 2, children in high-character class had less bullying and bullied behavior than those in low-character class (Bullying: χ^2^
_(1)_ = 119.96, *p* < 0.01; Bullied behavior: χ^2^
_(1)_ = 149.53, *p* < 0.01). Children in low-character class had more bullying and bullied behavior than those in average-character class (Bullying: χ^2^
_(1)_ = 25.80, *p* < 0.01; Bullied behavior: χ^2^
_(1)_ = 86.27, *p* < 0.01). Children in high-character class had lower bullying and bullied behavior than those in average-character class (Bullying: χ^2^
_(1)_ = 66.82, *p* < 0.01; Bullied behavior: χ^2^
_(1)_ = 62.30, *p* < 0.01).

### 4.3. The Moderating Effect of Left-behind Status on the Association of Moral Character with Children’s Bullying and Bullied Behavior

MANCOVA was performed using SPSS21.0 to examine whether the associations of moral character with bullying and bullied behavior differed between LBC and NLBC after age and gender were controlled for. A new interaction term (left-behind status × latent character classes) was created. The results indicated that the interaction was not significantly correlated with bullying behavior (*F*_(2, 653)_ = 0.535, *p* > 0.05, η^2^ = 0.002) but was significantly correlated with bullied behavior (*F*_(2, 653)_ = 3.517, *p* < 0.05, η^2^ = 0.011).

To help interpret the significant interaction, we graphed the interaction. As illustrated in Figure 2, NLBC in low-character class had the highest score on bullied behavior (*M* = 0.77, *SD* = 0.73) than those in high-character class (*M* = 0.26, *SD* = 0.47) and average-character class (*M* = 0.61, *SD* = 0.72) did. However, for LBC, there was no significant difference between low-character class and average-character class; LBC in high-character class scored lower on bullied behavior than those in low-character class.

## 5. Discussion

This study uniquely contributes to the existing literature by documenting the association of the latent character classes with bullying and bullied behavior in a sample of children from rural China, especially including LBC and NLBC. Additionally, we attempted to contribute to the current knowledge by focusing on the moderating effect of left-behind status on the relationship of the latent character class with children’s bullying and bullied behavior. Such contributions will allow practitioners and policymakers to draw references from the literature when designing interventions or preventions for children’s bullying and bullied behavior. The present study used LPA to identify the latent moral character classes. Children’s moral character was divided into three classes: low-character class, average-character class, and high-character class, respectively. Then the research was carried out based on the LPA results, which found significant grade and gender differences in the latent character classes. However, there were no significant differences between LBC and NLBC in the latent character classes. In addition, significant differences in bullying and bullied behavior were found among different classes of moral character, and left-behind status moderated the effect of latent character class on children’s bullied behavior.

What are the characteristics of moral children? Our results indicated that they were considerate of others, good at self-control, and grateful. Specifically, they were able to consider the views and feelings of others (high empathy, low uncaring) and refrain from manipulating others (low Machiavellianism). In addition, they could resist temptation (high self-control) and have a grateful heart (high gratitude). Moreover, they were generally described as disciplined, prudent, and organized (high conscientiousness).

Our results also indicated that children’s latent moral character could be divided into three classes (i.e., low-character class, average-character class, and high-character class). This study also found that children in the low-character class were more likely to be involved in bullying and bullied behavior, while children in the high-character class were less likely to bully others or be bullied. Previous studies have confirmed the relationship between personality traits and bullying. For example, people who lacked empathy [45], had a low level of conscientiousness, tended to control others [21], and those who had low self-control [46] were more likely to participate in bullying. Moreover, gratitude was associated with children’s bullying behavior [47]. These results showed that children in the high-character class could consider the views and feelings of others, refrain from manipulating others, resist temptation, and had a grateful heart, making their living environment better. Therefore, we have reasons to believe that children in the high-character class are less likely to bully others or be bullied, while children in the low-character class are more likely to be the opposite. This may be because children with low character suffered more bullying and bullied behavior owing to their own maladaptive behavior, making their living environment worse. On the contrary, those in high-character class are more likely to protect others from bullying because they are compassionate and have a high sense of responsibility, which keeps them from being bullied by others.

What is more, this study further indicated that the relationship of the character classes with bullied behavior differed between LBC and NLBC. Specifically, for NLBC, children in the low-character class were more likely to be bullied than those in average-character and high-character classes. For LBC, children in the high-character class were less likely to be bullied than those in low-character and average-character classes, while children in low-character and average-character classes were both susceptible to being bullied. In China, most LBC were taken care of by their grandparents, and grandparents can easily spoil the children or fail to supervise them [14], which might make these LBC more likely to form bad habits and qualities, and further, more likely to have problem behaviors such as bullying and bullied behavior than NLBC [24]. LBC also suffer more bullied behavior due to their low character than NLBC with low character because they lack parental supervision, protection, and support [48]. LBC are a large group in rural China. We should pay more attention to this vulnerable group and develop character-based interventions to keep them away from negative events such as bullying and bullied behavior by cultivating their good character.

### 5.1. Limitations

Although this study has important theoretical and practical implications, there are several limitations that merit mention. First, this study is a cross-sectional study and cannot explain the causal associations between the interested variables. Future studies can use longitudinal or experimental designs to examine the complex relationship between children’s moral character and school bullying. Second, this study only included two types of school bullying: bullying and bullied behavior. It is necessary to classify the types of bullying more specifically, including physical, verbal, relational, and cyber-bullying or explore the profile of bullied behavior along with bullying behavior, such as non-involved, victims, bullies, and bullies–victims [49]. Third, this study examined the relationship of moral character with children’s bullying and bullied behavior. Parental supervision and parental and peer support could protect children away from school bullying [14]. Therefore, future research could explore the association of children’s character and ecological contexts (e.g., parental, teacher, and peer support) with child development.

### 5.2. Implications

This study has significant theoretical and empirical implications. First of all, this research revealed that children with high moral character had few bullying and bullied behaviors, which will have important implications for the prevention and intervention of school bullying. Second, this study found that LBC in the low-character class suffered more bullied behavior. Therefore, we need to pay special attention to these children to prevent school bullying. Finally, this study has important theoretical value. To the best of our knowledge, this study is the first to explore whether the association of moral character class with bullying/bullied behavior differed between LBC and NLBC. Until now, we knew little about LBC’s moral character, which would be detrimental to their positive development.

## 6. Conclusions

The present study found that children’s latent moral character can be divided into three classes (i.e., low-character class, average-character class, and high-character class), and children in the low-character class had more bullying and bullied behavior than those in average-character and high-character classes. Our findings also indicated that the association between moral character and bullied behavior differed between LBC and NLBC. Strategies for preventing bullying behavior should be explored because the prevalence of bullying is frequent among LBC, the proportion of LBC who occasionally suffered at least one type of school bullying was 49.2% [14,50]. This study demonstrated that high moral character could protect children from bullying and bullied behaviors. Effective policies and prevention programs are needed to consider students’ moral character to combat bullying behaviors in China.

## Figures and Tables

**Figure 1 ijerph-19-11285-f001:**
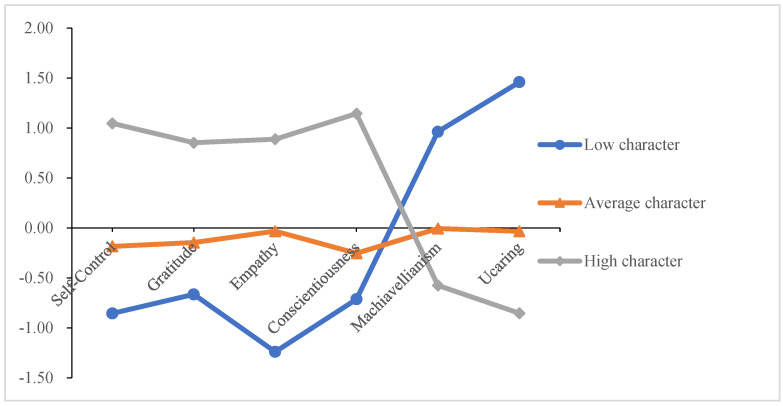
Moral character latent profile model.

**Figure 2 ijerph-19-11285-f002:**
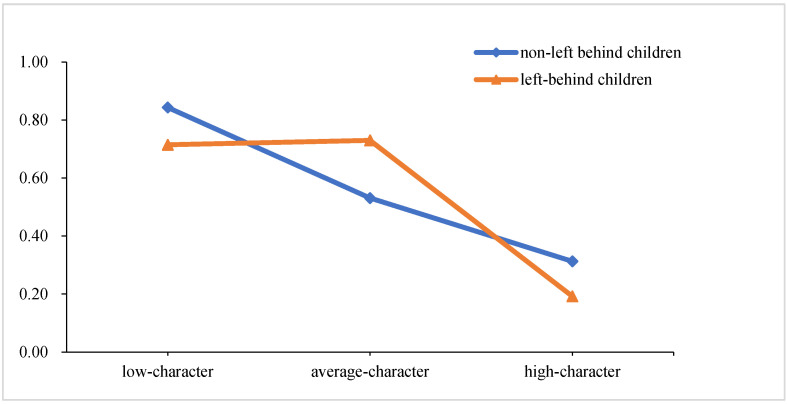
Interaction effect of left-behind status with latent character class on bullied behavior.

**Table 1 ijerph-19-11285-t001:** Model fit indices of LPA.

	AIC	BIC	LMRT	Entropy	Class Proportion
class 1	11,190.717	11,245.718	—	—	—
class 2	10,627.906	10,714.990	−5583.359 **	0.710	66.6/33.4
class 3	10,417.781	10,536.949	−5294.953 **	0.769	63.6/14.5/21.8
class 4	10,383.282	10,534.534	−5182.890 *	0.803	2.3/61.0/15.2/21.6

*Note*. * *p* < 0.05, ** *p* < 0.01.

**Table 2 ijerph-19-11285-t002:** The association of latent character classes with children’s bullying and bullied behavior.

	Low-Character Class		Average-Character Class		High-Character Class		Low vs. Average	Low vs. *High*	Average vs. High
	*M*	*SE*	*M*	*SE*	*M*	*SE*	χ^2^	χ^2^	χ^2^
Bullying behavior	0.58	0.04	0.31	0.02	0.05	0.02	25.80 **	119.96 **	66.82 **
Bullied behavior	1.92	0.13	0.65	0.03	0.30	0.02	86.27 **	149.53 **	62.30 **

*Note*. ** *p* < 0.01. χ^2^ refers to the cross-class mean equality test in the output results.

## Data Availability

Data are available from the first author upon request.

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
