# Peer review of "Latent Profile Analysis of Children’s Moral Character and the Classing Effect on Bullying in Rural China"

_ijerph, 2022, doi:10.3390/ijerph191811285_

Round 1

Reviewer 1 Report

1. It does not follow the format of citations and references established by Ijerph.
2. A paragraph is missing at the end of the theoretical framework explaining how the article has been organized, the subject that it seeks to address, preliminary information, discussion and conclusion.
3. Justify starting hypotheses and alternative hypotheses.
4. What type of scope is intended to be carried out in the research? Descriptive or correlational? Please justify in the text.
5. A series of questionnaires are mentioned, but the application procedure is not known. When? All questionnaires at the same time? Was it carried out in phases? In general, the methodology should be extended in order to understand the results obtained.
6. To deepen the discussion regarding the hypothesis raised.

Author Response

Thank you very much for giving us an opportunity to revise our manuscript. We appreciate editor and reviewers very much for the positive and constructive comments and suggestions on our manuscript entitled “Latent Profile Analysis of Children’s Moral Character and the Classing Effect on Bullying in Rural China”. (ID: ijerph-1819904).

We have studied reviewers’ comments and suggestions carefully and have made revision which marked in red in the paper. We have tried our best to revise our manuscript and hope that the revision will meet with approval. Attached please find the revised version, which we would like to submit for your kind consideration.

We would like to express our great appreciation to you and reviewers for comments on our manuscript. Looking forward to hearing from you.

Thank you and best regards.

Reviewer #1: Comments and Suggestions for Authors.

  1. It does not follow the format of citations and references established by Ijerph.

Answer: We are so sorry for our negligence that we did not follow the format of citations and references established by Ijerph, we have revised the format in the manuscript, for example:

However, due to the restrictions of China’s household registration system and financial constraints, migrant parents have to leave their children behind in the hometown to be looked after by others [1].

  1. Lu, Y., Zhang, R., & Du, H. Family structure, family instability, and child psychological well‐being in the context of migration: Evidence from sequence analysis in China. Child Development, 2021, 92, E416–E438. https://doi.org/10.1111/cdev.13496

  1. A paragraph is missing at the end of the theoretical framework explaining how the article has been organized, the subject that it seeks to address, preliminary information, discussion and conclusion.

Answer: Thank you much for pointing this out. In the manuscript, we have explained the theoretical basis of the article in different places. And we have integrated them together and revised the manuscript.

P1, paragraph 2: “Different from the deficit model, Positive Youth Development (PYD) emphasized the developmental potential of adolescents themselves rather than incompetence [8]. PYD was primarily concerned with three areas of children’s development: the nature of the child; the interaction between the child and the community; and moral growth [6]. Among these three areas, the role of moral character was prominent that may be more uniformly and globally associated with positive outcomes early in development [13].”

P2, paragraph 4: “According to the relational developmental systems theory, moral character in-volves linking across time and place to provide mutually positive benefits to both self and others [20]. Enhancing children’s character could benefit both individuals and civil society [21]. Complementary to the relational developmental systems theory is the dyadic agent-patient model of morality, which proposes that harmful acts are commit-ted by moral agents and these acts cause suffering to moral patients [11].”

P3, paragraph 2: “According to PYD, LBC have the potential for good development. Character is a core element of PYD [20]. And the relational developmental systems theory indicates moral character could provide mutually positive benefits to both self and others [21]. In accordance with these perspectives, the criterion variables used in this study are children’s bullying and bullied behavior. However, only a few studies have examined specific character traits in LBC and the associations with bullying and bullied behavior. Until now, no research has ever directly examined the possibility that the association of moral character with bullying and bullied behavior differs between LBC and NLBC.”

P3, paragraph 3: “We selected some specific traits of moral character by searching for abundant literatures on LBC and conducted latent profile analysis (LPA) to determine which measures best distinguished individuals with low moral character from those with high moral character. This person-centered method could capture all information at the individual level [30]. Based on the above theoretical foundation and empirical studies, this study firstly explored the potential latent classes of children’s moral character. Next, we explored the relationship of moral character with children’s bullying and bullied behavior. Finally, we investigated whether left-behind status moderated the association of moral character with children’s bullying and bullied behavior. We postulated that children’s latent moral character could be divided into three classes (i.e., low-character class, average-character class, and high-character class), and children’s moral character was correlated with bullying and bullied behavior, and left-behind status could moderate the association of moral character with children’s bullying and bullied behavior.”

  1. Justify starting hypotheses and alternative hypotheses.

Answer: Considering the reviewer’s good suggestion, we have revised the manuscript.

P3, paragraph 2: “According to PYD, LBC have the potential for good development. Character is a core element of PYD [20]. And the relational developmental systems theory indicates moral character could provide mutually positive benefits to both self and others [21]. In accordance with these perspectives, the criterion variables used in this study are children’s bullying and bullied behavior.”

P3, paragraph 3: “We selected some specific traits of moral character by searching for abundant literatures on LBC and conducted latent profile analysis (LPA) to determine which measures best distinguished individuals with low moral character from those with high moral character. This person-centered method could capture all information at the individual level [30]”

  1. What type of scope is intended to be carried out in the research? Descriptive or correlational? Please justify in the text.

Answer: We are so sorry for our negligence that we did not report the type of results. We have revised the manuscript.

P5, paragraph 3: “First, descriptive statistic was used to examine children’s latent moral character. LPA was implemented in Mplus7.0 to distinguish moral character profiles based on Z scores of these six character traits (i.e., self-control, gratitude, empathy, conscientiousness, Machiavellianism, and uncaring). Model fit was based on the Akaike information criterion (AIC), Bayesian information criterion (BIC), Entropy, and Lo–Mendell–Rubin likelihood ratio test (LMR-LRT) [30, 45]. Low scores of these indices showed a good fit to the data. Entropy indicated model classification accuracy, and if Entropy was more significant than 0.80, the model classification accuracy exceeds 90%. LMRT was used to compare the model, and a significant value (p < 0.001) indicated the k model is better than the k-1 model. Theoretical interpretability of the classes was considered in comparing models with similarly good fit statistics. Second, mixed regression analysis and multivariate analysis of covariance were used to analyze the association of children’s latent moral character with bullying/bullied behavior and the possibility that this association differs between LBC and NLBC.”

  1. A series of questionnaires are mentioned, but the application procedure is not known. When? All questionnaires at the same time? Was it carried out in phases? In general, the methodology should be extended in order to understand the results obtained.

Answer: We are so sorry for our negligence that we did not report the application procedure. We have revised the manuscript.

P3, paragraph 5: “This study was approved by the research ethics committee of our institution. The researchers obtained the informed consent of parents and participants before data collection. The participants were assured that they were free to withdraw and their responses would be kept confidential. We designed student questionnaire 1, student questionnaire 2, and parent questionnaire to conduct this investigation. Student questionnaire 1 was used to obtain children’s demographic information and character traits. Student questionnaire 2 was used to obtain information of children’s bullying and bullied behavior. In this study, we did not use information reported by parents. The students were asked to complete student questionnaire 1 and student questionnaire 2 in their classroom during different class sessions that lasted approximately 30 min. The researchers explained the requirements and instructions during the survey in class-rooms and guided the participants to ensure that they correctly understood the questionnaire.”

  1. To deepen the discussion regarding the hypothesis raised.

Answer: Considering the reviewer’s good suggestion, we have revised the manuscript.

P7, paragraph 2: “The present study used LPA to identify the latent moral character classes. Children’s moral character was divided into three classes: low-character class, average-character class, and high-character class, respectively. Then the research was carried out based on the LPA results, which found significant grade and gender differences in the latent character classes. However, there were no significant differences between LBC and NLBC in the latent character classes. In addition, significant differences in bullying and bullied behavior were found among different classes of moral character. And left-behind status moderated the effect of latent character class on children’s bullied behavior.”

P8, paragraph 1: “This may be because children with low-character suffered more bullying and bullied behavior owing to their own maladaptive behavior, making their living environment worse. On the contrary, those in high-character class are more likely to protect others from bullying because they are compassionate and have a high sense of responsibility, which keeps them from being bullied by others.”

P8, paragraph 2: “In China, most LBC were taken care of by their grandparents, and grandparents could easily spoil the children or fail to supervise them in place [16], which might make these LBC more likely to form bad habits and qualities and further more likely to have problem behaviors such as bullying and bullied behavior than non-left-behind children [26]. And Left-behind children would suffer more bullied behavior due to their low character than non-left-behind children with low character because they lack parental supervision, protection, and support [50]. Left-behind children are a large group in rural China. We should pay more attention to this vulnerable group and develop character-based interventions to keep them away from negative events such as bullying and bullied behavior by cultivating their good character.”

Reviewer 2 Report

We would like to thank the authors for the state of the art presented in this work. The subject is extremely interesting and new.

The introduction is concise and with a good description of the topic and the problems

We propose two aspects to consider:

·        On section number 2 Back ground it would be of interest to introduce some theoretical framework that will refound the investigation. Given the topic to be addressed in this study, we must consider the existence of numerous theoretical currents that support the proposed results.

·        On point 3 Methods, it would be important to introduce how the schools included in the research were selected.

Author Response

Thank you very much for giving us an opportunity to revise our manuscript. We appreciate editor and reviewers very much for the positive and constructive comments and suggestions on our manuscript entitled “Latent Profile Analysis of Children’s Moral Character and the Classing Effect on Bullying in Rural China”. (ID: ijerph-1819904).

We have studied reviewers’ comments and suggestions carefully and have made revision which marked in red in the paper. We have tried our best to revise our manuscript and hope that the revision will meet with approval. Attached please find the revised version, which we would like to submit for your kind consideration.

We would like to express our great appreciation to you and reviewers for comments on our manuscript. Looking forward to hearing from you.

Thank you and best regards.

Reviewer #2: We would like to thank the authors for the state of the art presented in this work. The subject is extremely interesting and new. The introduction is concise and with a good description of the topic and the problems.

  1. On section number 2 Back-ground it would be of interest to introduce some theoretical framework that will refound the investigation. Given the topic to be addressed in this study, we must consider the existence of numerous theoretical currents that support the proposed results.

Answer: Considering the reviewer’s good suggestion, we have revised the manuscript.

P1, paragraph 2: “Previous literatures were mainly based on the deficit model and focused on the problems and potential risks that LBC face [3, 4]. For example, compared with non-migrant children, LBC had increased risk of depression, anxiety, suicidal ideation, conduct dis-order, substance use and stunting [5]. Some researchers argue that if a child had developmental deficits in childhood that were not directed and corrected during their critical periods of development, those deficits would be difficult to change in adulthood [6, 7]. Different from the deficit model, Positive Youth Development (PYD) emphasized the developmental potential of adolescents themselves rather than incompetence [8]. PYD was primarily concerned with three areas of children’s development: the nature of the child; the interaction between the child and the community; and moral growth [6]. Among these three areas, the role of moral character was prominent that may be more uniformly and globally associated with positive outcomes early in development [13].”

P2, paragraph 4: “According to the relational developmental systems theory, moral character involves linking across time and place to provide mutually positive benefits to both self and others [20]. Enhancing children’s character could benefit both individuals and civil society [21]. Complementary to the relational developmental systems theory is the dyadic agent-patient model of morality, which proposes that harmful acts are committed by moral agents and these acts cause suffering to moral patients [11].”

P3, paragraph 3: “We selected some specific traits of moral character by searching for abundant literatures on LBC and conducted latent profile analysis (LPA) to determine which measures best distinguished individuals with low moral character from those with high moral character. This person-centered method could capture all information at the individual level [30].”

  1. On point 3 Methods, it would be important to introduce how the schools included in the research were selected.

Answer: We are so sorry for our negligence that we did not report how the schools included in the research were selected. We have revised the manuscript.

P3, paragraph 4: “Data were collected from 8 rural primary and middle schools in Henan province, China, including 4 primary schools and 4 middle schools. In 2018, there were approximately 699,000 LBC in Henan Province, accounting for 10.1% of the total number of LBC in the country. These eight schools were all from economically underdeveloped areas of Henan Province, a region of midland China with a substantial proportion of migrant labor. The current data collected is therefore rather representative of the general LBC in China.”

Round 2

Reviewer 1 Report

Changes where ´properly done.